# Can Schwartz Center Rounds support healthcare staff with emotional challenges at work, and how do they compare with other interventions aimed at providing similar support? A systematic review and scoping reviews

Cath Taylor,[1,2] Andreas Xyrichis,[2] Mary C Leamy,[2] Ellie Reynolds,[2] Jill Maben[1,2]

[1]School of Health Sciences, University of Surrey, Guildford, UK
[2]Florence Nightingale Faculty of Nursing, Midwifery & Palliative Care, King's College London, London, UK

**Correspondence to**
Dr. Cath Taylor;
cath.taylor@surrey.ac.uk

## ABSTRACT

**Objectives** (i) To synthesise the evidence-base for Schwartz Center Rounds (Rounds) to assess any impact on healthcare staff and identify key features; (ii) to scope evidence for interventions with similar aims, and compare effectiveness and key features to Rounds.

**Design** Systematic review of Rounds literature; scoping reviews of comparator interventions (action learning sets; after action reviews; Balint groups; caregiver support programme; clinical supervision; critical incident stress debriefing; mindfulness-based stress reduction; peer-supported storytelling; psychosocial intervention training; reflective practice groups; resilience training).

**Data sources** PsychINFO, CINAHL, MEDLINE and EMBASE, internet search engines; consultation with experts.

**Eligibility criteria** Empirical evaluations (qualitative or quantitative); any healthcare staff in any healthcare setting; published in English.

**Results** The overall evidence base for Rounds is limited. We developed a composite definition to aid comparison with other interventions from 41 documents containing a definition of Rounds. Twelve (10 studies) were empirical evaluations. All were of low/moderate quality (weak study designs including lack of control groups). Findings showed the value of Rounds to attenders, with a self-reported positive impact on individuals, their relationships with colleagues and patients and wider cultural changes. The evidence for the comparative interventions was scant and also low/moderate quality. Some features of Rounds were shared by other interventions, but Rounds offer unique features including being open to all staff and having no expectation for verbal contribution by attenders.

**Conclusions** Evidence of effectiveness for all interventions considered here remains limited. Methods that enable identification of core features related to effectiveness are needed to optimise benefit for individual staff members and organisations as a whole. A systems approach conceptualising workplace well-being arising from both individual and environmental/structural factors, and comprising interventions both for assessing and improving the well-being of healthcare staff, is required.

## Strengths and limitations of this study

► This is the first systematic review of Schwartz Center Rounds (Rounds), a healthcare staff intervention from the USA that has spread rapidly through UK healthcare organisations.

► Additional scoping reviews of 11 interventions with similar aims to support the well-being of healthcare staff, enables a novel comparative analysis to key features of Schwartz Rounds.

► This paper compares other staff well-being interventions to Rounds, thereby resulting in a focus on key features of Rounds; we did not explicitly draw out key features of other interventions or compare them against each other.

► The use of scoping reviews for comparator interventions, and exclusion of evidence in populations other than healthcare staff means that some evidence may have been omitted.

► The heterogeneity of study designs and outcomes, and weak study designs, means that findings are summarised narratively rather than using meta-analysis.

Schwartz Rounds could be considered as one strategy to enhance staff well-being.

## INTRODUCTION

In this paper, we report the systematic review of evidence regarding Schwartz Center Rounds (Rounds) and conduct a comparative analysis of 11 interventions also broadly aimed at supporting healthcare staff with the emotional challenges of their work. In doing so, we define Rounds from the literature and discuss the future potential use of interventions to support staff with the emotional challenges of providing healthcare. Healthcare providers are among the largest employers in

many countries worldwide. For example, the UK National Health Service (NHS) employs 1.5 million staff,[1] and in 2014 there were approximately 1.8 million physicians,[2] and 3.4 million nurses[3] across the European Union. Provision of healthcare relies on both clinical and non-clinical staff (eg, managers, administrators, porters/orderlies, caterers and domestic staff), all of whom may be impacted by the emotional challenges they face in their interactions with the patients and families they come across in day-to-day life.

Numerous publications have highlighted the high prevalence of psychological morbidity among healthcare staff in both clinical and non-clinical roles, and in many different countries worldwide.[4–10] Indeed, studies have typically reported between a quarter to a third of healthcare staff to have levels of psychological distress indicative of the need for clinical intervention, and in the UK mental health reasons explain a third of all NHS sickness absence, costing approximately £1 billion (of the total £2.4 billion cost of sickness absence in 2015).[11] Together with the clear consequences of this for their well-being and quality of life, and impact on their families, there is now increasing recognition of the link between the well-being of healthcare staff and quality of patient care (in relation to both patient experience and clinical outcomes).[12–15]

Consequently, the well-being of healthcare staff is high on the agenda of healthcare organisations in the UK and worldwide.[16–21] In the UK, the National Institute for Health and Care Excellence guidance published in 2009 recommended that organisations take a strategic approach to tackling staff well-being, encompassing approaches that focus on both prevention and treatment and that include interventions for individuals as well as 'organisation-wide approaches that encompass all employees'.[22] However, the reviews underpinning this guidance highlighted the poor quality of evidence overall and in particular the limited evidence on organisation-wide policies or approaches, with the strongest evidence in relation to interventions aimed at stress management for individuals.[22]

Schwartz Rounds are a rare example of an organisation-wide intervention that has seen rapid spread across healthcare organisations in the UK.[23] Rounds originated in the USA where they now run in over 430 organisations. After a pilot introduction to two UK hospitals in 2009, they now run in over 170 UK health and social care organisations (hospitals, hospices, community settings). They were developed to support healthcare staff to deliver compassionate care by providing a safe space where staff could openly share and reflect on the emotional, social and ethical challenges faced at work. The premise is that caregivers will be more able to make personal connections with colleagues and patients if they have insight into their own responses and feelings. Their rapid adoption in the UK was despite a limited evidence base, although attendance at Rounds was reported to be associated with improved compassion for patients, better teamwork and reduced stress in staff members, as well as having a positive impact on organisational culture.[24 25]

Consequently, the National Institute for Health Research commissioned a national evaluation of Rounds that has recently concluded,[26] supporting these earlier findings and showing attendance at Rounds to be associated with a reduction in psychiatric morbidity. A key component of the evaluation, intended to support organisational decision-making regarding staff well-being interventions, was to review the evidence for Schwartz Rounds and contextualise them by comparing the features of Rounds to other staff well-being interventions with similar aims. This paper reports the results from this, and thereby aims to answer the following review questions:

1. What are the defining features of Schwartz Center Rounds, and what is their evidence base?
2. What comparable interventions providing staff support/reflective space exist, what key features do they share with Schwartz Rounds and what is their evidence base in healthcare professionals?

Specifically, we aim to (i) identify key features of Rounds by synthesising published descriptions of Rounds to create a composite definition; (ii) systematically review and appraise all empirical evaluations of Rounds; (iii) identify comparative interventions, describe their key features and scope their evidence base and (iv) document similarities and differences between Rounds and comparative interventions.

## METHODS

The review of Schwartz Rounds literature followed Preferred Reporting Items for Systematic Reviews and Meta-Analyses systematic literature review guidance where applicable.

### Search strategy

The search strategies for the systematic review of Rounds literature involved: (i) a traditional database search (PsychINFO, CINAHL, MEDLINE and EMBASE to give comprehensive coverage of medical, psychological, nursing and social sciences literature). As an example, the MEDLINE database search for Schwartz Rounds was: (Schwartz adj2 Round*).mp. (mp=title, abstract, original title, name of substance word, subject heading word, keyword heading word, protocol supplementary concept word, rare disease supplementary concept word, unique identifier), (ii) use of internet search engines and (iii) consultation with experts. Inclusion criteria included having a health professional sample (either qualified or trainee) and empirically evaluating the intervention using qualitative and/or quantitative methods. The review excluded non-English language sources, unpublished dissertations/theses and any papers not accessible via the institution's online library, Google Scholar or directly from the journal website. All records were pooled together into a bibliographic database. First, records were screened to exclude duplicate entries. Second, the title and abstract of remaining records was reviewed for eligibility. All database searches were conducted between 14

October 2014 and 5 February 2015, although searches for Schwartz Rounds evaluations and consultation with experts continued until September 2017.

### Data extraction and quality appraisal

Standard data items were extracted to describe included papers (eg, citation, country, setting, population/sample, overall design, etc) and the evaluation (eg, length of evaluation; data collection method/s; outcome measures; key findings) using extraction sheets that were developed and piloted by all data extractors. In addition, items were developed that were specific to each intervention, for example, whether group or individual focused, size of group, length/number of sessions, content of sessions, whether facilitated or not (and if facilitated whether training/supervision was provided). Quality assessment of qualitative and quantitative primary studies was undertaken for each study using the tools developed by Jones *et al*,[27] which include assessment of key criteria and then an overall rating (high—no or few flaws; moderate—some flaws; low—significant flaws). Mixed methods studies were, in addition, assessed against the six criteria for good reporting of mixed methods studies developed by O'Cathain *et al*.[28] Quality was rated low (<3 criteria were met); moderate (3–4) or high (5+).

### Synthesis

Thematic analysis of the types of outcomes reported resulted in the identification of three categories relating to: a) self; b) others (eg, patients, colleagues) or c) wider organisation (eg, changes to policies; organisational metrics such as safety or satisfaction). Findings are presented according to these three categories. Finally, the overall quality of the evidence base for each intervention is described based on the range in quality for individual studies.

### CONSTRUCTING A COMPOSITE DEFINITION OF SCHWARTZ ROUNDS

While the Schwartz Center for Compassionate Care, where Rounds originated, have a description of Schwartz Rounds on their website, this was found to omit key aspects of their design that we knew from scoping the literature to be important (eg, an ongoing programme, time-fixed in length, food is provided, it is open to all staff and panellists stories are preprepared). Therefore, we constructed a 'composite' definition based on descriptions used in Rounds literature in order to determine the key features of Rounds for comparison with other interventions. For this process, we included all literature (including non-empirical literature, eg, letters, editorials) providing it included a description of Rounds. Text describing Rounds (what they were and their intended aims, eg, structure and purpose, as well as any text describing what they were 'not') was extracted from published accounts. The text was analysed thematically by four team members independently (CT, JM, ML, MH),

core concepts were discussed and agreed and a single definition was produced. The face validity of the definition was confirmed after review by study advisory and steering group members.

### SCOPING REVIEWS OF COMPARATIVE INTERVENTIONS

The reviews of comparable interventions followed an interpretative scoping literature review methodology based on the framework outlined by Arksey and O'Malley.[29] The searching, data extraction and synthesis followed similar steps to the review of Schwartz Rounds literature (except where noted below) but instead of producing a detailed critique and review of individual studies they were instead aimed at producing a summary description of the evidence base in relation to size, scope and quality, and used to extract data relevant for the comparative analysis. For each intervention, the number/type of included papers was recorded, and each intervention was described in relation to its original format (eg, number of participants, original setting and healthcare setting/s and intended aims/outcomes); and the variability in its application within the literature (fidelity to original format). Main findings were examined across all interventions and analysed thematically (using the same categories as for Schwartz Rounds: self, others, organisation) to enable synthesis within, and comparison across, each intervention.

### Identification of comparative interventions to include

We aimed to identify interventions that support health professionals with the emotional challenges of delivering patient care. Initially, we identified aspects that were fundamental to Rounds, including providing an opportunity for reflection, disclosure and offering psychological safety; and these informed choices regarding potential comparative interventions. Included interventions needed to focus on psychological (as opposed to physical) well-being of staff; be person-directed (vs work directed) and provide primarily emotional rather than cognitive/clinical support (eg, excluding mortality/morbidity meetings, which aim to provide lessons in terms of cognitive errors or systems issues). Although Rounds are a 'group' (rather than individual) intervention, we chose not to limit comparative interventions by this characteristic, due to the importance of reflection and/or disclosure as a key potential mechanism in Rounds that is shared by other interventions that are not group-based. Potential comparative interventions were identified through published reviews of psychological/emotional support interventions for healthcare staff[30–33] and through consultation with steering and advisory group members (with expertise in Rounds/well-being interventions in healthcare). A total of 11 interventions were scoped: action learning sets; after action reviews; Balint groups; caregiver support programme; clinical supervision; critical incident stress debriefing; mindfulness-based stress reduction;

peer-supported storytelling; psychosocial intervention training; reflective practice groups; resilience training.

## Comparative analysis to Schwartz Rounds

The composite definition of Rounds was disaggregated into its individual descriptive features which were extracted into a table, together with the features that were 'not' part of Rounds. Further clarification was added for some descriptive features to ensure clarity of meaning (eg, 'reflection' became 'provides an explicit opportunity for reflection'). The description of each comparative intervention was then reviewed by the research team and assessed in relation to whether or not it also provided each of the key features of Rounds. The face validity of the comparison between Rounds and other interventions was confirmed with study advisory and steering groups (with expertise in Rounds/healthcare staff well-being interventions).

## PATIENT INVOLVEMENT

We actively involved patients through membership of the Project Steering Group (PSG), which included two patient public involvement (PPI) representatives (Havi Carel, Christine Chapman) who had previously provided input to the original funding application. The PSG provided oversight to all aspects of the study, and alongside other group members our PPI representatives and Rounds staff members advised on design, inclusion of comparative interventions and commented on the findings.

## RESULTS
### Key features of Rounds

Forty-three documents/sources were included in the review of descriptions of Rounds (table 1), which allowed development of the definition.[24 25 34–74] The majority (n=33) were non-empirical publications (eg, commentaries, descriptive reports of a single Round). The thematic synthesis resulted in the production of the composite definition (see online supplementary file 1), a summary version is provided in table 1.

## EVIDENCE BASE FOR ROUNDS: RESULTS FROM THE SYSTEMATIC REVIEW

Twelve empirical evaluations of Rounds were included (table 2) arising from 10 studies (4 in the USA, 6 in the UK). Most were mixed methods evaluations, typically comprising attenders completing evaluation forms post-Round attendance, followed by interviews or focus groups (n=5), one mixed method study comprised case studies (observation/interviews) together with descriptive analysis of evaluation forms[75] and one used both quantitative and qualitative methods to analyse evaluation forms.[76] Two were quantitative studies, and one qualitative study. Only one study included non-attenders[66] (table 2).

Overall quality of the evidence-base was assessed to be low/moderate. Most studies had study designs prone to risk of bias (eg, cross-sectional), used non-validated questionnaires (typically self-report views/satisfaction with Rounds and impact of attendance) and none of the quantitative evaluations had control group (non-attender) comparisons. Little information was provided on the samples/sampling frames in quantitative studies (eg, in relation to breadth of professional group representation or role in Rounds), nor were findings analysed or presented in relation to such factors. In two studies that did report the characteristics of their quantitative sample, most were female and of white ethnicity, and nurses predominated (but neither study reported the seniority of nurse).[24 47] Findings from these studies included that Rounds are highly valued by attenders (although represented a small proportion of total staff). Most studies reported positive impact on 'self' (eg, improved well-being, coping)[24 25 44 47 49 51 59 66 70 75 76] and impact on patients (increased compassion, empathy)[24 25 51 59 66 70 75 76] and colleagues (improved teamwork, compassion/empathy).[24 25 44 47 50 51 59 66 75 76] Six studies provide evidence of wider institutional impacts from interviews with attenders[24 25 44 51 66 75 76] (table 2). Three of the included studies were evaluations of Rounds adapted for educational purposes[39 50 70]; all reporting that Rounds were felt to be useful and that students gained knowledge/understanding about the emotional side of providing patient care.

## COMPARATIVE INTERVENTIONS: RESULTS FROM THE SCOPING REVIEWS

Electronic searches for the 11 comparative interventions yielded a total of 1725 papers, of which 146 were included (ranging between 1 and 64 across interventions, table 1, see online supplementary file 2 for included references). A number of publications (n=253) were not obtainable due to being published in sources that no longer existed or not available through institutional subscription and internet searches. The largest evidence base was for clinical supervision (n=64) followed by Balint groups (n=26). Half of the studies were quantitative (n=74: RCT, observational, quasi-experimental), 41 were qualitative (mixed designs, interviews, focus groups), 22 were mixed methods and 9 were secondary studies (literature reviews). The literature was international with the majority of studies from the USA and the UK; other countries represented included Canada, Australia, Finland, Norway, Sweden, Croatia, Spain, Italy, Israel and South Africa. There was a distinct lack of studies from Asia, although that may be a reflection of the English language limit.

For most interventions, high-quality evidence was sparse. Populations for many of the interventions lacked diversity across health professions and settings, with many mostly nursing-focused. The aims of studies varied widely, with a few aimed at assessing efficacy or effectiveness but most were

**Table 1** Search results for each intervention

| Intervention | Description of intervention | Database search result | Total excluded (duplicates; not eligible; full paper not available) | Papers from experts/ internet search | Total number included for review |
|---|---|---|---|---|---|
| Schwartz Center Rounds (Rounds) | Regular (usually monthly) open forum for staff in all positions. Organised and managed by a steering group, championed by a senior clinician and facilitated—usually by a senior doctor and psychosocial practitioner. Last for 1 hour, with food provided. Multiple perspectives on a theme, scenario or patient case are briefly presented by a pre-arranged and preprepared panel and then opened to the audience for group reflection and discussion. Focus on non-clinical aspects (psychosocial, ethical, emotional issues) surrounding the patient-caregiver relationship (see online supplementary file 1 for a full definition). | 41 | 0 | 2 | 43 (33 non-empirical; 10 empirical) |
| Action learning sets (ALS) | Based on the concept of learning by reflection on (or reviewing) an experience, ALS usually contains 4–6 members (peers), with (or without) a 'set advisor' to facilitate the process. ALS tend to be held intermittently, over a fixed programme cycle, and most participants contract with the facilitator for an agreed length of time. They are often closed groups. The set is not a team, as the focus is on actions of individuals, rather than shared work objectives. | 83 | 70 (8; 36; 26) | 1 | 14 |
| After action reviews (AAR) | AARs are facilitated meetings, led by a senior member of staff, which aim to encourage active reflection on performance following a specific event. An AAR is a one-off meeting postevent and includes those who were involved with the event. The focus is on gaps in performance, and what could be done differently to enhance the outcome. AARs generally last about 30 min. | 76 | 74 (9; 64; 1) | 0 | 2 |
| Balint groups | Balint groups meet every 1–4 weeks for 1–3 years. In the group, typically a doctor presents a troubling patient incident while the group listens. The goal of the presentation is to understand the issue from both the patient's and doctor's perspectives. The presentation can last about 10 min, after which group members can ask clarifying questions. When all questions are exhausted, the group is invited to imagine themselves in the roles of the doctor and the patient. | 384 | 358 (170; 151; 37) | 0 | 26 |
| Caregiver support programme (CSP) | Originally developed for mental health/learning disability care homes, CSP is described as a theory-based social support intervention aimed at increasing exchanges of social support and decreasing negative social interaction. It consists of six 4–5 hour group training sessions (10 managers, 10 direct-care staff and 2 facilitators) conducted over a 9-week period. Strategies for improvement are drawn from the participants, based on their own experiences. | 84 | 83 (10, 71, 2) | 2 | 3 |

Continued

**Table 1** Continued

| Intervention | Description of intervention | Database search result | Total excluded (duplicates; not eligible; full paper not available) | Papers from experts/ internet search | Total number included for review |
|---|---|---|---|---|---|
| Clinical supervision | Clinical supervision originated in psychotherapy and adopted by other disciplines, eg, psychology/nursing. Process described as identifying a key issue, describing and defining it, undertaking a critical analysis, examining solutions, formulating an action plan, implementation and evaluation. It can take five different forms: one-to-one with expert from same discipline; one-to-one with supervisor from different discipline; one-to-one with colleague of similar expertise; supervision between groups of colleagues working together and network supervision between people who do not usually work together. | 307 | 252 (42, 160, 50) | 9 | 64 |
| Critical incident stress debriefing (CISD) | In its original form, CISD is a single-issue debriefing session in a group context, led by an external team, following a traumatic event. CISD has seven phases: *introduction, fact, thought, reaction, symptom, teaching* and *re-entry*. The debriefing session lasts for approximately 1.5–3 hours and takes place 24–72 hours after the traumatic event. The debriefing team is made up of a leader, a co-leader and a support, who work in conjunction to support the participants and to allow them to feel safe. | 388 | 386 (62; 248; 76) | 0 | 2 |
| Mindfulness-based stress reduction (MBSR) | The central principle of MBSR is mindfulness—being focused on and aware of the present moment with a non-judging attitude of acceptance. The original training module is 8 weeks long with weekly sessions of 2.5 hours each. There is a 7-hour session, which takes place between weeks 6 and 7, and participants are asked to complete 45 min of daily formal mindful practice. They are taught a variety of mindful meditative practices, and there are group discussions about the application of these practices. | 127 | 110 (13; 72; 25) | 0 | 17 |
| Peer-supported storytelling | Narrative storytelling is the act of an individual recounting verbally to one or more people a plausible account of an event, or series of events, possessing narrative truth for the teller. The story is arranged in a time sequence with plot, characters, context, intentionality and perspective taking, possibly including the teller's actions, thoughts and feelings. | 4 | 3 (3; 0; 0) | 0 | 1 |
| Psychosocial intervention training | Psychosocial intervention training involves cognitive behavioural approaches for managing symptoms, understanding symptom-related behaviour, relationship formation and helping service users to cope with symptoms. Teaching sessions are supplemented by small group supervision. Students are required to provide brief case study presentations about service users they are working with and receive feedback. Early courses were developed for nurses but quickly became multidisciplinary. | 37 | 35 (6; 25; 4) | 1 | 3 |

Continued

**Table 1** Continued

| Intervention | Description of intervention | Database search result | Total excluded (duplicates; not eligible; full paper not available) | Papers from experts/ internet search | Total number included for review |
|---|---|---|---|---|---|
| Reflective practice groups (RPG) | RPGs are facilitated groups of about 10 healthcare professionals or students in which participants share and explore professional, clinical, ethical and personal insights arising from their clinical work or training. RPGs are ongoing, convening regularly with each group lasting for about 1 hour. Discussion topics can either be raised by the facilitator or by the participants. The discussion is meant to be supportive as well as challenging, encouraging consideration of alternative viewpoints. | 91 | 83 (8; 73; 2) | 0 | 8 |
| Resilience training | Resilience training is in part based on CBT theories and in its original form is a manualised intervention comprising 18 hours of workshops. The key characteristics include delivery to groups of practitioners who support one another and are facilitated by an expert in personal and professional transition supervision. University of Pennsylvania well-known example consists of: learning ways to challenge unrealistic negative beliefs, strengthening problem solving, adopting assertiveness and negotiation skills, improving ability to deal with strong feelings and learning how to tackle procrastination through use of decision-making and action-planning tools. | 144 | 138 (36, 72, 30) | 0 | 6 |
| Total | | 1725 | 1592 (367; 972; 253) | 13 | 146 |

**Table 2** Schwartz Rounds empirical evaluations: data extracted from included papers

| Authors | Setting | Aims/purpose | Design/methodology | Measures | Main findings | Quality |
|---|---|---|---|---|---|---|
| Corless et al [39] | Educational USA | Development, implementation and evaluation of Educational Rounds for interdisciplinary graduate students to help them learn empathy, self-reflection and moral courage. | Quantitative post-Round evaluation survey. Graduate students. Over 4 years comprising 11 Rounds (n=329 individual evaluations). | Survey included seven statements about Rounds (according to agreement on a 5-point Likert scale) plus an overall rating of the quality of the Round they are evaluating. | *Overall:* High support and satisfaction with Rounds (eg, 86% rated Rounds as excellent or exceptional). 67% stated intention to attend future Rounds (range 57%–93% for individual Rounds). Lowest intention from a Round presented by lab scientists. Highlighted importance of topic to encourage attendance. | Low/moderate (lack of clarity, eg, sampling, measures, not all data presented). |
| Manning et al [59] Lown and Manning [24] | Hospitals USA | To assess the impact of Rounds, eg, changes in attendees behaviours and beliefs about patient care, teamwork, stress and personal support. | Mixed method evaluations ▲ Retrospective survey of 256/413 (62%) attenders at six experienced Rounds sites (offering Rounds for 3+ years) plus at 44 interviews with providers, Rounds leaders, facilitators and hospital administrators. ▲ Prospective prepost web-based survey of 222/399 (56%) Rounds attenders from 10 hospitals newly implementing Rounds (had held seven or more Rounds) | Study-specific (non-standardised)—some adapted from published measures) Likert scale measures to investigate: 1) Insights into psychosocial and emotional aspects of clinical care on patient interactions (15 items), 2) Teamwork (nine items). 3) Support for providers (number of items not mentioned). | *Overall:* Found 'close' effect: more Rounds attended, more impact they have. *Self:* Attendance at Rounds associated with decreased stress and improved ability to cope with psychosocial demands/emotional difficulties at work. *Others:* Rounds attendance led to increased patient interaction and teamwork scores. Interviews highlighted benefits including: getting to know colleagues and putting themselves in their shoes, and an improved sense of connection/shared purpose. *Organisation:* Both samples (51% retrospective; 40% prospective) reported changes in practices/polices including: culture change (dialogue that does not happened elsewhere); focus on patient-centred care; practice changes (eg, increased/earlier palliative care use). | *Quantitative:* Moderate (lack of control group of non-attenders and used non-standardised measures). *Qualitative:* Moderate (limited reporting of theoretical underpinnings and strategies to improve rigour, eg, deviant case analysis). *Mixed methods:* Low (limited reporting of mixed methods rationale/ features). |
| Goodrich [51] Goodrich [25] | Two hospitals UK | Pilot study to evaluate introduction of Rounds to the UK in two hospital sites. | Mixed methods evaluation over 2-year period: 1) Prepost pilot surveys. 2) Evaluation forms from each Round (quality, logistics, demographics, plans to attend future Rounds). Each site held 10 Rounds (n=301 attenders site A, 74% completed evaluation form; n=949 at site B, 69% completed evaluation form). 3) Qualitative interviews: experience of attenders, steering group, panellist, facilitators (n=23). Second interview at end with n=13. | Used same questionnaires as Lown and Manning.[24] | *Overall:* Majority (86% site A, 78% site B) rated rounds as excellent/good. *Self:* Increases (pre post) in: ▲ confidence in handling sensitive issues; ▲ beliefs in the importance of empathy. ▲ Confidence in handling non-clinical aspects of care. Also reported feeling less stressed and less isolated in their work. Interview findings: increased compassion, reduced stress. *Others:* Increases (prepost) in: ▲ actual empathy with patients; ▲ openness to expressing thoughts, questions and feelings about patient care with colleagues. Interview findings: greater respect/empathy for colleagues, better teamwork/collaboration. *Organisation:* Interview findings: ▲ Board/senior support important; ▲ Wider impacts: reduced hierarchy, help build shared values/support strategic vision. | *Quantitative:* Moderate (limitations in measures used and lack of control group). *Qualitative:* Moderate (low reporting of strategies to improve rigour and theoretical underpinnings). *Mixed methods:* Low (limited reporting of mixed methods rationale/ features). |

**Table 2** Continued

| Authors | Setting | Aims/purpose | Design/methodology | Measures | Main findings | Quality |
|---|---|---|---|---|---|---|
| Reed et al[66] | Hospice UK | Evaluate the impact of Rounds on staff and the organisation. | Longitudinal mixed methods evaluation (1 year): survey and focus groups. Exit survey: 398/535 (74%) attendees. Four interprofessional focus groups (n=33, including attendees, non-attendees and presenters). | 5-Point Likert scale assessing: topic relevance, knowledge gained, impact on individual, facilitation and working relationships. (Similar questions to Lown and Manning[24]/Goodrich[51]). | *Overall:* 78% rated Rounds as excellent or exceptional. *Self:* ▲ Focus groups; themes included:Validation of experiences;Honesty, openness and vulnerability allowed others to see person on human level. *Others:* 87% gained insight into how others think/feel in caring for patients: ▲ Focus groups: fostered understanding of importance of non-clinical staff contribution. BUT non-attenders felt responsibility to smooth running of hospice and felt they contributed to wider team without needing to hear stark realities of care/work. *Organisation:* ▲ Focus groups: more connected, shared purpose. | *Quantitative:* Moderate, (non-validated measures and lack of control group). *Qualitative:* Moderate (lack of elements of rigour in qualitative component, eg, reflexivity, contradictory/deviant cases other than non-attenders). *Mixed methods:* Low (limited reporting of mixed methods rationale/ features). |
| Deppoliti et al[44] | Hospital USA | ▲ Learn why people attend Rounds. ▲ Understand what is gained from the experience. ▲ Identify key elements to use in measuring effectiveness. | Qualitative: four focus groups (n=27) and three telephone interviews. Purposive sampling of attenders by steering group to represent those that were active contributors and included range of roles/professions and frequency (low and high attenders). | NA | *Overall:* Rounds viewed as beneficial. *Self:* ▲ Impact on behaviour/attitudes 'think differently'. *Others:* ▲ Exposing emotions (increased appreciation, awareness and sensitivity of what others in the healthcare team experience). ▲ Walking in another's shoes (empathic awareness). *Organisation:* ▲ Culture change (strong message that staff matter; values/beliefs/norms evolved positively; not about productivity; improved teamwork due to level playing field). *Other findings:* ▲ Inequality of topics (some topics more than others lead to increased learning, growth). ▲ Influence of rules and boundaries (spoken/ unspoken rules about what is acceptable to share *Suggested improvements:* ▲ Providing list of upcoming topics so staff can plan attendance. ▲ Providing anonymised method to contribute (eg, Qs on cards). | *Qualitative:* High (good reporting and transparency in methods of data collection and analysis including, eg, independent researcher coding/analysis, third-party review of findings and use of illustrative quotations). |

Continued

**Table 2** Continued

| Authors | Setting | Aims/purpose | Design/methodology | Measures | Main findings | Quality |
|---|---|---|---|---|---|---|
| George[47] | Hospital UK | To examine the impact of Schwartz Rounds on staff well-being and patient care. | *Mixed methods:* Interviews with staff (nurses and HCAs) about stress (n=11, 10 were female, 10 were white British). Key themes extracted using grounded theory → development of a new measure administered at the beginning and end of 2 Rounds (n=55 forms completed). Mostly female, white and only 2 were over 59 years. | The Organisational Response to Emotions Scale (ORES) (investigator-designed): nine scales. Analysis controlled for whether it was first ever Round, length of time in role, session attended. | *Self:* ▲ Emotional labour: significantly reduced in staff where preround was their first round. ▲ Self-reflection increased prepost. ▲ Compared Rounds attenders with 10/11 interviewees who also completed ORES (did not attend Rounds). Found non-attenders had higher burnout and emotional labour, and more negative appraisal of organisation. *Others:* - More negative appraisal of line manager. | *Quantitative:* Moderate (small biased sample, lack of control group, measure based on limited staff group input (nurses/HCA only). *Qualitative:* Moderate (limited reporting of elements of rigour, eg, audit trail, theoretical saturation). *Mixed methods:* High (good reporting of mixed methods). |
| Shield et al[70] | Medical school USA | To improve communication skills, they designed 'Schwartz Communication Sessions'. Aimed to provide medical students with the rationale and proficiency for effective communication. | Quantitative: evaluation form (both quantitative and qualitative/content analysis). Sampling is unclear (a sample of 92, 99 and 94 are reported, but report having evaluation forms from 66% to 95% of students for all three sessions (71%–80% for two sessions) and faculty members (n=24) response rate 42%–92%. | Not specified (but appear similar to Lown and Manning).[24] | *Overall:* 93% of faculty and 83% of students rated the sessions as good, excellent or exceptional. *Self:* 80% of students and 96% of faculty believe students gained knowledge that will help them care for patients. *Others:* 75% of students and 96% of faculty believe the sessions will help students communicate better with patients and family members. | Moderate (lack of clarity regarding sampling/sample and measures). |
| Gishen et al[50] | Medical school UK | Examine the potential of Rounds within the undergraduate curriculum. | *Mixed method evaluation:* Two student-focused Rounds were piloted at a medical school (1 Round each for year 5 and 6 students). Evaluation questionnaire immediately following the Rounds: 258/334 (77%) year 5 students attended the Round and 247 (94%) responded. 180/343 (52%) year 6 students attended the Round and 126 (70%) responded. Focus group (n=7 year 5 students) to explore student views on the Round. | Feedback form from the Point of Care Foundation; plus free-text comments. Questions either yes/no or 5-point Likert rating scale (1=poor to 5=exceptional). | *Overall:* Mean student ratings of a session were 3.5/5 (year 5) and 3.3/5 (year 6): ▲ 81% agreed/strongly agreed the presentation of cases was helpful. ▲ 80% would attend a future Round. ▲ 64% agreed Rounds should be integrated into the curriculum. *Focus group findings:* Feelings about the Round (response to Round, size of audience—large inhibiting, positive comparison to current reflective practice; postevent peer discussions). *Self:* ▲ 69% year 5 vs 87% year 6 students were worried about compassion fatigue or burnout. ▲ 92% agreed/strongly agreed that they appreciated hearing stories demonstrating human side of medicine. ▲ Focus group finding: psychological aspects of Rounds (psychological pressures of medicine, how session encouraged positive processing of emotion, sharing personal stories between health professionals). *Others:* 82% agreed/strongly agreed that attending Round gave insight into how others feel/think about caring for patients. | *Quantitative:* Moderate due to convenience sampling approach and lack of control group. *Qualitative:* Moderate due to limited reporting of measures taken to enhance rigour *Mixed methods:* Low for mixed methods reporting. |

Continued

**Table 2** Continued

| Authors | Setting | Aims/purpose | Design/methodology | Measures | Main findings | Quality |
|---|---|---|---|---|---|---|
| Farr and Baker[75] | Community and mental health services UK | To investigate how Schwartz Rounds are implemented and how they support staff in mental health and community services. | *Mixed methods:* Realist evaluation involving three case studies including 22 interviews, 5 observations and descriptive analysis of evaluation sheets as secondary data. Key themes extracted using framework analysis, then used to track context, mechanism outcome configurations. | Topic guides developed from a previous evaluation of Schwartz Rounds (Goodrich[25]) Process data collected through evaluation sheets, ie, attendance, types of participants and perspectives. | *Overall:* Rounds had the potential for successful implementation in these settings. *Self:* More mindful of emotional impact of work. *Others:* Positive impact on relationships with patients and other staff (eg, better communication, increased trust). *Organisation:* Rounds can positively influence wider culture of the organisation (eg, prompt introduction of other forms of staff support). *Other findings:* Facilitators and barriers to implementation are identified including that strong leadership is a crucial element in success. | *Qualitative:* Moderate (methods for realist methods and analysis lack clarity, eg, unclear presentation of Context-Mechanism-Outcome configurations) *Quantitative:* Low (descriptive data on attenders only; lack of clarity on analysis of evaluation sheet data and integration with interview data). *Mixed methods:* Low (limited reporting of mixed methods rationale/features). |
| Chadwick et al[76] | Hospital UK | To assess the perceived impact of Schwartz Rounds on hospital staff over a 3-year period. | *Mixed methods analysis of evaluation forms:* 795 participant evaluation forms analysed quantitatively and qualitatively. Key themes extracted using thematic analysis. Quantitative (scale) data analysed descriptively and inferentially (eg, for differences in scores for different items/scales). | Standard evaluation form, eight statements rated on 5-point scale and free-text comments. | *Overall:* All aspects of Rounds highly rated by staff. *Self:* Improved reflection on own experiences. *Others:* Improved insight and understanding of others and their roles and support within the organisation. Most agreed that attending the Rounds helped caring for patients and work with colleagues. *Organisation:* Reducing the effect of 'silo working'. Staff were given a space where powerful emotions were accepted and responded to constructively. Need to make Rounds more accessible to wider cross-section of staff. *Other findings:* ▲ No significant differences between disciplines/staff groups in survey ratings. | *Qualitative:* High. *Quantitative:* Moderate (statistical methods unclear; lack of account of impact of sample size on outcomes of analysis). *Mixed methods:* Not a mixed methods study, just in analysis of evaluation data, therefore criteria NA. |

HCA, Healthcare Assistant; NA, not applicable.

small-scale exploratory descriptive studies. The content and format of interventions (fidelity) was in most cases widely heterogeneous (and/or lacked detail), and consequently synthesis of findings is problematic. Most of the quantitative evaluations across all interventions relied on weaker study designs (eg, cross-sectional studies, postintervention evaluations, lacking control comparisons), used non-probability sampling, had small samples likely to be underpowered and used non-validated outcomes measures. Many qualitative studies also lacked clear reporting of aspects of rigour (eg, limited reporting of member checking, deviant cases, reflexivity or evidence of data saturation). A summary of the evidence base for each intervention is provided in online supplementary file 3.

### Synthesis

Most interventions presented evidence in relation to all three categories of outcomes ('self', 'others' and 'organisation'), although evidence for resilience training, mindfulness-based stress reduction and reflective practice groups lacked inclusion of organisational outcomes. All of the interventions had evidence of positive benefits to self (eg, raised self-awareness, resilience, job satisfaction, empowerment or overall well-being), and most provided some evidence of positive benefits to 'others'. Impact on patients included fostering of better provider-service user relationships, communication with and/or attitudes towards patients and improved patient-centredness, knowledge of patients' suffering and empathy. Impact on colleagues, included associations with better teamwork, peer support and knowledge/understanding of colleagues.

At organisational level, there was evidence from some interventions of association with improved practice, for example, reductions in unnecessary prescriptions, increased uptake of psychosocial support (Balint groups), reduction in task and coordination errors and increased uptake of postfall huddles (after action reviews). Two interventions provided evidence of a positive impact on the workforce, including providing opportunities for mentoring and advice (action learning sets) and improved staff retention (clinical supervision).

### SCHWARTZ VERSUS ALTERNATIVE INTERVENTIONS: COMPARATIVE FEATURES

In comparison to the other interventions, Rounds offer a unique organisation-wide 'all-staff' forum to share stories about the emotional impact of providing patient care (table 3). While many of the other interventions expect 'open, honest communication' as a key feature, and provide an explicit opportunity for reflection, none is open to all staff (eg, clinical and non-clinical, voluntary attendance) and many are not ongoing programmes but instead are one-off training courses or events. Some of the training interventions (eg, mindfulness-based stress reduction, or resilience training) are multidisciplinary in training attendance, but conduct/practice of the intervention occurs subsequently and is individual, compared

with Schwartz Rounds (and other interventions such as Balint groups), where learning and practice take place simultaneously in group settings.

Other key aspects in which Rounds are distinct from the comparative interventions relate to what Rounds are intentionally 'not' meant to be. In particular, discussions within Rounds should not 'problem solve' in order to avoid focus on the clinical decision-making in a patient case, whereas problem solving/action planning are key features of many of the other interventions (eg, action learning sets, after action reviews, critical incident stress debriefing). Most of the comparative interventions also offered flexibility in format, compared with Rounds which require a contractual licence (with stipulated conditions) obtained via the Schwartz Center for Compassionate Care (USA) or Point of Care Foundation (PoCF, UK).

Arguably the closest types of interventions to Rounds are Balint groups (although rooted in unidisciplinary primary care—physicians only—with closed membership), and reflective practice groups (again generally closed membership and can be unidisciplinary). In particular, both are ongoing group programmes in which challenging/rewarding experiences about delivering patient care are shared and discussions are facilitated, and both provide the opportunity to give and/or receive peer support in safe and confidential environments. However, neither offers an organisation-wide opportunity for staff to attend, and both would have an expectation that members/attenders would contribute, whereas in Rounds attenders can choose to be silent listeners. Clinical supervision can also provide an opportunity to reflect on the emotional and ethical challenges of care without problem-solving/action planning, but unlike Rounds this usually occurs in a one-to-one situation, not group, and requires those being supervised to verbally contribute.

### DISCUSSION

Our work revealed a rich portfolio of available interventions to support staff with the emotional challenges of providing healthcare, each designed with different audiences and uses in mind. The evidence base regarding the effectiveness of these largely remains weak, and more should be done to examine these more systematically. The studies reviewed here show some evidence of impact at different levels, and future work should seek to unpick which interventions work best, under which conditions and for which participants. To our knowledge, this is the first comparative review of staff well-being interventions.

Given the high rates of work-related stress and mental health issues among healthcare staff, it is not acceptable for employers not to act, despite the weak evidence base for most approaches and interventions currently. Some staff groups have clinical supervision, for example, as an integral part of their work (mental health nurses; midwives; psychologists and social workers), whereas most doctors and nurses do not, and such staff often have little or no support with the emotional, social and

**Table 3** Features of Rounds compared/contrasted with comparative interventions

| Feature of Rounds | | Intervention (see footnote for full label) | | | | | | | | | | |
|---|---|---|---|---|---|---|---|---|---|---|---|---|
| | | ALS | AAR | Balint | Care | Super | CISD | Mind | Story | Psych | Refl | Resil |
| 1 | Share challenging/rewarding experiences about delivering patient care | May Not | No | Yes | May Not | Yes | Yes | No | May Not | Yes | Yes | May Not |
| 2 | Focus on psychosocial and emotional issues of patient-caregiver relationships | May Not | No | Yes | May Not | May Not | May Not | No | May Not | May Not | May Not | May Not |
| 3 | Provides an explicit opportunity for reflection | Yes | Yes | Yes | May Not | Yes | Yes | Yes | Yes | Yes | Yes | May Not |
| 4 | Open, honest communication | Yes | Yes | Yes | Yes | Yes | Yes | Yes | Yes | Yes | Yes | May Not |
| 5 | Provides an opportunity to give and/or receive peer support | Yes | May Not | Yes | Yes | Yes (if group) | Yes | Yes | Yes | No | Yes | Yes |
| 6 | Telling and hearing stories related to a theme, scenario or patient case | No | Yes | No | May Not | No | No | No | Yes | No | No | No |
| 7 | Ongoing programme (vs one-off) | No | No | Yes | No | Yes | No | No | No | No | Yes | No |
| 8 | Time-fixed session (vs flexible length/unspecified) | No | No | Yes | Yes | Yes | No | Yes | No | Yes | No | Yes |
| 9 | Planned provision of food/refreshments | No | No | No | No | No | No | No | No | No | No | No |
| 10 | Open to all/any clinical and non-clinical staff | No | Yes | No | No | No | No | No | No | No | No | May Not |
| 11 | All levels of staff/intended to flatten hierarchy | Yes | Yes | No | Yes | No | No | Yes | Yes | Yes | No | Yes |
| 12 | Open group membership (vs closed/invited members only) | No | No | No | No | No | No | No | No | No | No | No |
| 13 | Multidisciplinary | May Not | Yes | May Not | Yes | No | May Not | Yes | May Not | Yes | No | May Not |
| 14 | Preprepared/rehearsed stories or focus | Yes | No | No | No | No | No | No | Yes | No | No | No |
| 15 | Facilitated discussions | May Not | Yes | Yes | Yes | Yes | Yes | Yes | No | Yes | Yes | Yes |
| 16 | Panel presenters tell stories giving their perspectives on a theme, scenario or patient case | No | No | No | No | No | No | No | No | No | No | No |
| 17 | Group intervention | Yes | Yes | Yes | Yes | May Not | Yes | Yes | No | Yes | Yes | Yes |
| 18 | Organisational support: senior doctor/clinician champions | May Not | Yes | Yes | Yes | Yes | May Not | No | No | No | No | No |
| 19 | Safe and confidential environment | Yes | Yes | Yes | Yes | Yes | Yes | Yes | Yes | Yes | Yes | Yes |

| Features that define what Rounds are 'not' | | Intervention | | | | | | | | | | |
|---|---|---|---|---|---|---|---|---|---|---|---|---|
| | | ALS | AAR | Balint | Care | Super | CISD | Mind | Story | Psych | Refl | Resil |
| 1 | Problem-solving | Yes | Yes | Yes | Yes | Yes | May Not | No | No | Yes | No | Yes |
| 2 | Production of actionable outputs | Yes | Yes | No | No | May Not | No | No | No | No | No | No |
| 3 | Flexibility in format (vs licensed/contract* | Yes | Yes | Yes | May Not | Yes | Yes | Yes | Yes | No | Yes | May Not |

**Table 3** Continued

| Features that define what Rounds are 'not' | | Intervention | | | | | | | | | | |
|---|---|---|---|---|---|---|---|---|---|---|---|---|
| | | ALS | AAR | Balint | Care | Super | CISD | Mind | Story | Psych | Refl | Resil |
| 4 | Focus on clinical aspects of patient care, their diagnosis or plan of care | May Not | May Not | No | May Not | May Not | May Not | No | May Not | Yes | May Not | May Not |

*Licensed/contract—fidelity to original intervention, ie, one model/approaches or many, degree of flexibility offered.
AAR, after action reviews; ALS, action learning sets; Balint, Balint groups; Care, caregiver support programme; CISD, critical incident stress debriefing; Mind, mindfulness-based stress reduction; Psych, psychosocial intervention training; Refl, reflective practice groups; Resil, resilience training; Story, peer-supported story-telling; Super, clinical/restorative supervision.

ethical challenges of their work. Non-clinical staff—who may have much contact with patients and the events they encounter—are even more neglected in relation to the impact of delivering patient care on them. Selection of interventions should be based on a strategic approach that incorporates needs assessment, implementation of interventions/approaches and policies and monitoring and review to determine the impact of these and refine/revise as necessary. There is a need for a range of approaches, not a one-size-fits-all and our work does not suggest an either/or approach for individual interventions. Rounds should not be seen as a replacement for or instead of clinical supervision (or other support/interventions), but could be offered to staff in addition. Organisation-wide interventions are important to tackle workplace environmental/cultural factors that impact on well-being; to change attitudes and cultural norms around staff needing support as well as changing conversations in organisations around empathy, compassion and the support required to deliver these. Involving all employees may improve coworker and supervisor support, which in turn can facilitate the development of a supportive workplace environment that reduces stress by improving attitudes and behaviours.[77] Compared with other interventions reviewed here, Rounds offer a unique organisation-wide 'all staff' forum to reflect on the emotional impact of providing patient care, offering opportunities for staff to reflect, whether or not they choose to disclose/contribute to discussions, and accruing evidence suggests they may have many benefits to individuals, others (colleagues, patients) as well as wider organisational impacts.[26]

Schwartz Rounds were originally conceived to meet a very specific identified need in healthcare: to support healthcare providers to be compassionate to patients through giving them insight into their own thoughts, feelings and behaviours.[26] In the UK, the reasons given for adoption has been more about staff well-being, in line with evidence linking quality of patient care and experience with staff well-being.[13 23] Unlike many of the other interventions, they have a structured format, and are specifically not intended to be 'problem-solving'. In doing so, they provide a 'counter-cultural' space that differs from the protocol-driven, outcome-orientated healthcare environment that values emotional stoicism: 'Good Rounds shift an organisation and its workers away from their default position of urgent

action, reaction and problem solving to an hour of stillness and slowness'[78] (p. 41). A key ingredient supporting Rounds to meet their intended aims is good facilitation, thus the role of the facilitator is key. Unlike the facilitation role in other interventions we reviewed, where there was often much variability in relation 'fidelity', in the UK, it is mandatory for Rounds facilitators to attend training provided by the PoCF (the UK licence holder for Schwartz Rounds), and they receive ongoing support from Schwartz mentors. It is recommended that there are at least two facilitators in each organisation, and the PoCF state that it helps if facilitators have experience of group work, and managing difficult emotions (many have psychology or social work backgrounds). In our national evaluation, we found despite most having these skills and background, they often shouldered the responsibility for Rounds on their own (some having only one facilitator too), which we found to impact negatively on their well-being, and on the sustainability of Rounds, recommending that a focus on facilitator support, and succession planning would be beneficial for Rounds[26 79]

Workforce interventions are often complex in nature, with many components and aims. Their evaluation is thereby challenging, particularly with regard to attributing any changes to outcomes to the intervention as opposed to other causes within the organisation/system. The challenge of conducting a robust evaluations of organisation-wide interventions may be one explanation as to why such evidence is so sparse,[22] and for why there is instead a predominance of evidence regarding individually targeted interventions such as mindfulness-based stress reduction. The application of new methodologies to address these challenges, such as realist evaluation, could enable a more robust understanding of how and why interventions work (or do not work), and has recently been applied in the first UK national evaluation of Schwartz Rounds.[26]

### Limitations

The focus of this review on the evidence within healthcare staff meant that wider evidence for some interventions, beyond healthcare, was not considered. Also, the scoping methods applied to the comparable interventions inevitably means that some relevant evidence may have been omitted, although systematic electronic searching and consultation with experts aimed to minimise this risk. The rapid uptake of Rounds in the UK and need to contextualise them within

staff well-being interventions, informed the design of this review. It was thereby a review that compared other interventions with the key features of Rounds, and did not thereby compare the key features of all the other interventions with each other, apart from by describing and synthesising their origins and evidence base.

## CONCLUSION

Given the time and resources already committed to the interventions considered here, it is important to determine how best to identify the core features of effectiveness to optimise benefit for individual staff members and organisations as a whole. This work has now been undertaken for Schwartz Rounds using a realist-informed methodology that has identified the contextual factors that influence how and for whom Schwartz Rounds work, resulting in an organisational guide giving practical guidance and recommendations for organisations to maximise the effectiveness of Rounds in their organisations.[26 79] The application of similar methodologies for other interventions such as clinical supervision and Balint groups may further help ensure optimal outcomes. A systems approach as opposed to an individual approach to tackling staff well-being, in order to improve patient care, is required, comprising effective interventions for assessing and improving the well-being of healthcare staff .

**Acknowledgements**  The authors would like to thank the members of the project advisory group (Joanna Goodrich, Barbara Wren, Jenny Firth-Cozens, Michael West, John Pepper, Glenn Robert) and steering group (Dean Fathers, Sharon Fleming, Nicola Mackintosh, Rachel Massey-Chase, Katherine Hopkins, Jude Bayly, Havi Carel, Christine Chapman, Elisabeth Buggins, Jennifer Gardner, Ruth Harris and Lisl Klein) for their guidance and comments throughout the study. The authors would also like to thank Michelle Hope for her contribution to the analysis of definitions of Schwartz Rounds.

**Contributors**  CT and JM conceived the original idea and designed the study; AX designed the search strategies and performed the database searches; CT, AX, ML and ER performed internet searches; screened titles and abstracts independently; extracted data and assessed quality for three interventions each; CT, JM, MCL and ER created the composite definition of Rounds. All authors contributed to the overall synthesis and comparative analysis; CT drafted the manuscript; AX, MCL, ER and JM revised critically the manuscript. All the authors approved the final version of the manuscript.

**Funding**  This work was supported by the National Institute for Health Research Health Services and Delivery Research Programme (project number 13/07/49).

**Disclaimer**  The views and opinions expressed therein are those of the authors and do not necessarily reflect those of the NHS, the NIHR, MRC, CCF, NETSCC, the HS&DR programme or the Department of Health.

**Competing interests**  JM reports that she was a member of an advisory group from 2006 to 2009, advising on the development of the Point of Care project at The King's Fund, and a member of the Point of Care Foundation Board 2013–2014; she stepped down as boardmember at the start of the evaluation. All other authors declare no conflicts of interest.

**Patient consent**  Not required.

**Provenance and peer review**  Not commissioned; externally peer reviewed.

**Data sharing statement**  No additional data are available.

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
