## [Reviewer comments · BMJ Open]

ARTICLE DETAILS

TITLE (PROVISIONAL)	Can Schwartz Center Rounds support healthcare staff with emotional challenges at work, and how do they compare to other interventions aimed at providing similar support? A systematic review and scoping reviews
AUTHORS	Taylor, Cath; Xyrichis, Andreas; Leamy, Mary C; Reynolds, Ellie; Maben, Jill

VERSION 1 – REVIEW

REVIEWER	Michelle Farr CLAHRC West, University of Bristol, UK
REVIEW RETURNED	28-Jun-2018

GENERAL COMMENTS	This is an excellent, comprehensive and detailed paper. It is very thorough in its content, providing a vast amount of detailed information in its analysis. It is very well researched and written, the authors have clearly outlined their systematic and comprehensive methods, and provided appropriate answers to research questions. In terms of how to improve the paper, I offer the following thoughts: 1. It would be helpful to explain what the benefit is of having a composite definition of Schwartz Rounds, in comparison with just taking the original format definition from the US Schwartz Centre. What are the aims and objectives of the authors in creating this composite definition? How can it inform future research?2. It is interesting that the word 'safe' is included in the composite definition of Schwartz Rounds (SR) (Supplementary file 1). In SR research interviews I have found that people's perceptions of a safe environment vary, and whilst the intention may be to create a safe environment, this may be a matter of context and perception. For example in one organisation where there was major restructuring and jobs were at stake, one interviewee spoke of how they may not feel safe to share their true feelings about their work, when there were senior managers present.3. Similarly to point 2, it may be intended that Rounds provide 'a level playing field', but whether they actually do or not is a matter of empirical investigation. Again, thinking back to research interviews, when I spoke to administrators they did not consider the SR discussion to be on a level.4. Therefore I wonder if in some places within the composite definition, the phrase 'it is intended' needs to be added in. When Schwartz Rounds are actually implemented in practice, some of these features may become difficult to clearly implement and evaluate i.e. what constitutes a safe and level playing field, and from whose perspective?5. Finally, associated with point 3, in Table 3 line 11, the Feature of Rounds "All level of staff/ no hierarchy" to me mean two very different things. All levels of staff may be invited to attend Rounds,
--

	and whilst it may be intended that all contributions are equally valued, this is different from having no hierarchy. Professional and medical hierarchies are culturally and discursively embedded, so to have 'no hierarchy' in these discussions may be difficult to facilitate and achieve. Changing the term 'no hierarchy' may be helpful here. Overall, a highly detailed, comprehensive and thorough study. Well done to the authors for their hard work.
--	--

REVIEWER	Dr. Shamsul Shah Auckland City Hospital, Auckland, New Zealand Chair of the Steering Group of Schwartz Center Rounds at Auckland City Hospital.
REVIEW RETURNED	01-Jul-2011

GENERAL COMMENTS	I congratulate the authors on researching and writing a worthy topic for discussion and challenging in its methodology. It is a question that is raised very often by clinicians and organisation leaders as to how Schwartz Rounds differ or benefit staff over and above other interventions e.g. Balint groups, individual supervision, critical incident groups etc. and so this paper offers and allows for discussion in this area. I liked that the authors conclude that there is 'no one size fits all' when it comes staff wellbeing and that an array of individual measures e.g. personal supervision, MBSR, team e.g. debriefs post events and organisation wide measures e.g. Schwartz Rounds can and should be considered to support staff wellbeing. There are some minor considerations only:  1. Language: there is an interchange of centre and center in the article and so consistency would be good. 2. Methods: PPI needs defining 3. Supplementary File 2: although there is a good evaluation of the studies conducted, I would have liked to have seen the numbers of patients or sample sizes for the comparator groups. 4. Table 2: Depolitti's study scored High in its quality although sample size was comparatively small and so would be good to detail as with the other scores as to why this was scored as such. 5. Methods: I do wonder whether a figure showing the scoring criteria for qualitative, quantitative and mixed-methods might be helpful but not essential. 6. Discussion: I wonder if there should be some mention of the importance of the facilitator role, formal training and experience in managing groups, internal versus external etc. as I note that this role can differ in the comparator interventions too.
---

VERSION 1 – AUTHOR RESPONSE

Reviewer: 1 (Michelle Farr)

This is an excellent, comprehensive and detailed paper. It is very thorough in its content, providing a vast amount of detailed information in its analysis. It is very well researched and written, the authors have clearly outlined their systematic and comprehensive methods, and provided appropriate answers to research questions.

Response: Thank you for this supportive and positive overview.

In terms of how to improve the paper, I offer the following thoughts:

It would be helpful to explain what the benefit is of having a composite definition of Schwartz Rounds, in comparison with just taking the original format definition from the US Schwartz Centre. What are the aims and objectives of the authors in creating this composite definition? How can it inform future research?

Response: The purpose of creating the composite definition was to determine the key features of Schwartz Rounds in order to provide a means for comparison to other interventions. Whilst the definition from the US Schwartz Centre for Compassionate Care contains many of these features, it was not as comprehensive as some of the definitions we found when we first scoped the literature, for example that it is an ongoing programme, time-fixed in length, that food is provided, that it is open to all staff, and that panellists stories are pre-prepared. By providing a more comprehensive definition it was possible to determine the elements that are similar and different to other interventions with similar aims. In relation to how the definition could inform future research, the creation of this definition informed the start of our work in our national evaluation to delineate the core (essential) and more peripheral (adaptable) elements of the intervention that should be 'measured' in order to assess for fidelity, and measure any variability in implementation. This work is relevant both to future evaluations of Schwartz Rounds and transferable to other similarly complex interventions.

Page 8 of the manuscript has been amended to say: "Whilst the Schwartz Centre for Compassionate Care have a description of Schwartz Rounds on their website, this was found to omit key aspects of their design that we knew from scoping the literature to be important (e.g. that it is an ongoing programme, time-fixed in length, that food is provided, that it is open to all staff, and that panellists stories are pre-prepared). Therefore, we constructed a 'composite' definition based on descriptions used in Rounds literature in order to determine the key features of Rounds for comparison with other interventions".

It is interesting that the word 'safe' is included in the composite definition of Schwartz Rounds (SR) (Supplementary file 1). In SR research interviews I have found that people's perceptions of a safe environment vary, and whilst the intention may be to create a safe environment, this may be a matter of context and perception. For example in one organisation where there was major restructuring and jobs were at stake, one interviewee spoke of how they may not feel safe to share their true feelings about their work, when there were senior managers present.

Response: Thank you for this comment. We agree that perceptions of safety will vary, and that this is an intended aim of Schwartz Rounds, and inbuilt to their design for example by having facilitators that have received training in running Schwartz Rounds, by preparing panellists beforehand to ensure their safety in sharing their story at that time, and by restating the rules regarding confidentiality at the start of every Rounds. We have amended the composite definition to clarify that this is the intended aim of Rounds. The features in this composite definition are taken from the definitions used in the literature – the majority of which used the word 'safe' as a defining feature of the Round.

Supplementary file 1 has been amended to include that this is intended.

3. Similarly to point 2, it may be intended that Rounds provide 'a level playing field', but whether they actually do or not is a matter of empirical investigation. Again, thinking back to research interviews, when I spoke to administrators they did not consider the SR discussion to be on a level.

Response: We agree with this, as with the comment above, and have amended the definition to make this clearer. Supplementary file 1 has been amended to include that this is intended.

Therefore I wonder if in some places within the composite definition, the phrase 'it is intended' needs to be added in. When Schwartz Rounds are actually implemented in practice, some of these features may become difficult to clearly implement and evaluate i.e. what constitutes a safe and level playing field, and from whose perspective?

Response: We agree and have amended the composite definition as suggested.

Finally, associated with point 3, in Table 3 line 11, the Feature of Rounds “All level of staff/ no hierarchy” to me mean two very different things. All levels of staff may be invited to attend Rounds, and whilst it may be intended that all contributions are equally valued, this is different from having no hierarchy. Professional and medical hierarchies are culturally and discursively embedded, so to have ‘no hierarchy’ in these discussions may be difficult to facilitate and achieve. Changing the term ‘no hierarchy’ may be helpful here.

Response: Thank you for this comment. Our meaning in relation to this feature in the table is not that all levels of staff can attend (as this is covered by the previous feature – open to all staff) but that hierarchies are not built into the design – that they are intended to give anyone and everyone a voice and to flatten hierarchies (unlike some other interventions where hierarchies are inbuilt to the design e.g. by senior clinicians leading groups). We have therefore changed the term in Table 3, line 11 to read ‘intended to flatten hierarchy’ to reflect this point.

Overall, a highly detailed, comprehensive and thorough study. Well done to the authors for their hard work.

Response: Thank you for your insightful and helpful comments.

Reviewer: 2 (Dr. Shamsul Shah)

I congratulate the authors on researching and writing a worthy topic for discussion and challenging in its methodology. It is a question that is raised very often by clinicians and organisation leaders as to how Schwartz Rounds differ or benefit staff over and above other interventions e.g. Balint groups, individual supervision, critical incident groups etc. and so this paper offers and allows for discussion in this area. I liked that the authors conclude that there is 'no one size fits all' when it comes staff wellbeing and that an array of individual measures e.g. personal supervision, MBSR, team e.g. debriefs post events and organisation wide measures e.g. Schwartz Rounds can and should be considered to support staff wellbeing.

Response: Thank you for your supportive summary of the manuscript.

There are some minor considerations only:

Language: there is an interchange of centre and center in the article and so consistency would be good.

Response: Thank you for spotting these – all references to Schwartz Center Rounds should have the American spelling (Center) so this has been corrected throughout. Patient-Centred appears a few times in the manuscript but has been left as the UK/English spelling.

Methods: PPI needs defining

Response: PPI is Patient Public Involvement. This has been defined in the manuscript.

Supplementary File 2: although there is a good evaluation of the studies conducted, I would have liked to have seen the numbers of patients or sample sizes for the comparator groups.

Response: The reviews of alternative interventions were scoping reviews, and as such “instead of producing a detailed critique and review of individual studies they were instead aimed at producing a summary description of the evidence base in relation to size, scope and quality, and used to extract data relevant for the comparative analysis” (as stated in the methods of the manuscript). These scoping reviews of the 11 interventions included a total of 146 individual papers covering a range of methodological approaches. The references for these are provided but it is not possible to include the

individual sample sizes for such a vast literature. In relation to 'comparator groups' only half of the studies were quantitative in design, and most did not have comparator groups, instead being pre-post designs.

Table 2: Depolitti's study scored High in its quality although sample size was comparatively small and so would be good to detail as with the other scores as to why this was scored as such.

Response: Depolitti's study was a qualitative study as such sample size is not relevant to assessment of quality. The omission of rationale for the rating given to this paper has been corrected. This scored high due to the high quality reporting and transparency in relation to methods of data collection and analysis, with the interpretation of findings clearly justified by the data presented. This manuscript included, for example, clear justification for sampling and method of data collection, transparent replicable method of data collection, and transparency regarding methods to enhance rigour for example having several researchers involved independently in coding/analysing the transcripts, and having a third party reviewer to examine the findings and choice of illustrative quotes. The table has been amended to include the following justification for the 'high' rating: good reporting and transparency in methods of data collection and analysis including e.g. independent researcher coding/analysis and third party review of findings and use of illustrative quotations.

Methods: I do wonder whether a figure showing the scoring criteria for qualitative, quantitative and mixed-methods might be helpful but not essential.

Response: We used the quality assessment methods used in a previously published paper in BMJ Open (reference below, including the link to the supplementary file where the criteria are presented) – these are freely available from the supplementary files for this previous study, though if the editors wished (and assuming permission from the authors of this work) we would be happy to also add them to this manuscript. For mixed methods studies we used criteria published by O'Cathain et al (reference below). We would not have permission to reproduce this though it is available via Open Athens and Shibboleth.

Jones CEL, Maben J, Jack RH, et al. A systematic review of barriers to early presentation and diagnosis with breast cancer among black women. *BMJ Open* 2014; 4:e004076. doi:10.1136/bmjopen-2013-004076 Supplementary file available here: <https://bmjopen.bmj.com/content/bmjopen/suppl/2014/02/12/bmjopen-2013-004076.DC1/bmjopen-2013-004076supp.pdf>

O'Cathain A, Murphy E, Nicholl J (2008) The quality of mixed methods studies in health services research. *J Health Serv Res Policy* 13(2):92-8. doi: 10.1258/jhsrp.2007.007074

Discussion: I wonder if there should be some mention of the importance of the facilitator role, formal training and experience in managing groups, internal versus external etc. as I note that this role can differ in the comparator interventions too.

Response: Thank you for this comment. We agreed that the role of the facilitator and how this is supported for Schwartz Rounds is very important, and have included the following text in the discussion:

A key ingredient supporting Rounds to meet their intended aims is good facilitation, thus the role of the facilitator is key. Unlike the facilitation role in other interventions we reviewed, where there was often much variability in relation 'fidelity', in the UK, it is mandatory for Rounds facilitators to attend training provided by the Point of Care Foundation (the UK Licence holder for Schwartz Rounds), and they receive ongoing support from Schwartz mentors. It is recommended that there are at least two facilitators in each organisation, and the PoCF state that it helps if facilitators have experience of group work, and managing difficult emotions (many have psychology or social work backgrounds). In our national evaluation, we found despite most having these skills and background, they often

shouldered the responsibility for Rounds on their own (some having only one facilitator too), which we found to impact negatively on their wellbeing, and on the sustainability of Rounds, recommending that a focus on facilitator support, and succession planning would be beneficial for Rounds [26, 79].